

# Brief communication -  Vent opening at Campi Flegrei: clues from dyke propagation patterns

Jacopo Selva[1], Nello Mangone[1]

[1] Dipartimento di Scienze della Terra, dell'Ambiente e delle Risorse (DiSTAR), University of Naples Federico II, Naples Italy

*Correspondence to*: Jacopo Selva (jacopo.selva@unina.it

**Abstract.** Forecasting future vent opening position is fundamental for managing volcanic hazards, and is usually based on the spatial density of past vents or other crust weakness indicators. Here, a novel empirical approach inspired by dyke propagation models is applied to the Campi Flegrei caldera. Results show that dyke direction correlates with topographic peaks within 6 km from the caldera center, and propagation length exhibits two main peaks at 2 and 4 km, leading to a vent opening probability map with maxima well correlating with recent major seismicity and deformations.

## 1 Introduction

The Campi Flegrei caldera volcanic activity dates back to the upper Pleistocene, with the oldest volcanic activity estimated at around 80,000 years ago (D'Antonio et al., 2007; Orsi et al. 2009). The first caldera collapse occurred approximately 39,000 years ago with the Campanian Ignimbrite (CI) eruption. The period following the CI is characterised by eruptions confined within the newly formed caldera, both continental and marine. A second major eruption occurred around 14,000 years ago, the Neapolitan Yellow Tuff (NYT), which likely caused a second collapse that shaped the current caldera structure (Sbrana et al., 2021). After the NYT eruption, volcanic activity resumed within the inner caldera (Orsi et al. 2004).

The post-NYT eruptive history is divided into three main epochs. The first epoch comprises a total of 33 eruptions, spaning from 15,000 to 10,600 years ago. The eruptive vents align with the newly formed caldera boundaries, with the most energetic eruption being that of the "Pomici Principali" around 12,300 years ago. The second epoch follows a relatively brief quiescent period and includes nine eruptions dated between 9,600 and 9,200 years ago, primarily concentrated in the northeastern sector of the caldera. The third epoch encompasses a total of 28 eruptions from approximately 5,500 to 3,800 years ago; this epoch is predominantly focused in the northeastern part of the caldera (Agnano area) and secondarily in the northwestern sector (Averno area). The most energetic eruption during this period was that of Agnano-Monte Spina around 4,550 years ago. The long period of quiescence following the third epoch lasted until 1538 AD, when the Monte Nuovo eruption took place in the northwestern sector of the caldera (Di Vito et al., 1999). The caldera has not experienced any eruption since.



After NYT, the caldera experienced significant resurgence, accompanied by seismicity, degassing, and slow ground
deformation often referred to as bradyseism. The latter is concentrated mainly in the geometrical center of the caldera, in the
Pozzuoli area, and major movements were tracked at least from the 4th century AD thanks to the ruins of a Roman temple in
the port of this city, known as the Serapeum. The largest known bradyseismic events are the ones related to the last Campi
Flegrei eruption (the 1538 Monte Nuovo eruption, Di Vito et al., 1999). In the last century, three main bradyseismic crises
occurred in 1950-52, 1970-72, and 1984-86, characterised by episodes of uplift of more than 1 m interrupting a slow
long-lasting subsidence (Del Gaudio et al 2010). Approximately in 2005, a slow uplift started, accelerating over time, which
now fully recovered the subsidence, exceeding the uplift peaks observed in the last century (Chiodini et al. 2021, Bevilacqua
et al. 2024, Giudicepietro et al. 2025). This event is still ongoing.
Forecasting vent opening is crucial for any volcanic hazard quantification. Different approaches were adopted at Campi
Flegrei through time. Alberico et al. (2002) developed a method identifying crustal weaknesses using geophysical,
geological, and geochemical parameters. Their probability map indicated the highest likelihood of vent openings in the
central caldera near Pozzuoli. Selva et al. (2012) utilized a Bayesian approach fed by fewer parameters, focusing on tectonic
structures to track crust weakness and past vents, shifting the area at higher probability toward the northeastern and
northwestern sectors, where post-NYT activity concentrated. Bevilacqua et al. (2015) adopted a method based on Gaussian
kernel and accounting for the uncertainty on past vent positions, confirming the northeastern sector (near Agnano and
Astroni) as the most likely area for future eruptions. More recently, based on past observations and removing multiple
eruptions from clustered vents, Charlton et al. (2018) noted that vent opening occurred substantially randomly within a ring
area surrounding the caldera center, corresponding to the NYT ring.
Rivalta et al. (2019) studied the physical propagation of magma dikes by modelling the trajectory of potential dikes due to
the subsurface stress field and the dike's initial position. Their model accounts for various stresses affecting magma ascent.
For calderas, it suggests that eruptive vents are concentrated at specific distances from the centre, influenced by the stress
induced by the caldera depression, defining a higher propensity to eruption closer to caldera rims, as noted by Charlton et al.
(2020). Considering the caldera depression size, Rivalta et al. (2019) forecasted for Campi Flegrei a potential peak for vent
opening at a semiannular belt located between 2.3 and 4.2 km from the caldera center. Rivalta et al. (2019) analyzed also the
effect of topographic peaks, which may break the caldera symmetry. They analyzed the case of the Campi Flegrei caldera,
explaining the concentration of volcanic activity in the northeastern sector due to the peak of the Camaldoli hill, which
creates a stress field favouring magma trajectories in the northeast direction.
The main features of Rivalta et al. (2019) model are i) that the geometry of the caldera significantly influences dike
propagation outward, promoting eruptions away from the geometric centre a a given distance, and ii) topographic
asymmetries create localised stress variations in the subsurface, affecting eruption frequency across different angular sectors.
While a sufficiently detailed knowledge of the sub-surface stress state is difficult to reach, it is possible to verify if these two
main features left track on the available record of past vent positions, and to use this empirical signature to define new vent
opening probability maps.





## 2 Empirical distribution of direction and length of past dykes

The path of the dykes feeding past eruptions is mainly controlled by the geometry of the caldera, which determines the distance from the centre of the caldera, and by the topographical peaks surrounding the caldera, which establishes preferential directions for propagation. Assuming that the origin at depth of the magma is located at the centre of the caldera, below the location of the maximum observed uplift (Amoruso et al. 2012, Rivalta et al. 2019), past vent positions may track the trajectories of the dykes that fed those eruptions. The centre can be approximately assumed at the point of maximum deformation, representing the centre of the bell-shaped deformation of Campi Flegrei. Here, it is set to LON 4263554 and LAT 451954 (UTM WGS84, zone 33N),

Based on this, it is possible to establish an empirical model for such trajectories by studying the empirical distribution of azimuth (angle to North) and distance-from-the-centre (hereinafter radius) of the caldera of the dykes that alimented past eruptions, and use this model to define a forecast map for future vents. Past vent positions are however affected by significant uncertainty. To account for this, we used the 71 vent positions and corresponding uncertainty bounds defined in Bevilacqua et al. (2015), accounting for the uncertainty by randomly sampling 1000 alternative synthetic positions within the defined uncertainty bounds (Supplementary Figure 1).

### 2.1 Length of dyke  propagation

At first, we analyse the distances between the centre of the caldera and all post-NYT vents, which represent the length of horizontal propagation of the dykes that alimented such eruptions. The analysis is conducted separately for the three epochs, and jointly for the entire dataset. For simplicity, the recent Monte Nuovo eruption is included in the third epoch. In Figure 1A, we report the empirical distributions (bins of 250 m, the corresponding empirical cumulative distribution functions are reported in Supplementary Figure 2), revealing a strong difference between the first epoch, where 60% of eruptions occur between 4,400 and 6,600 m, and the third epoch that shows shorter distances, with two significant peaks around 4,000 m and 2,000 m, being Epoch 2 somehow intermediate between the other two. A two-sample Kolmogorov-Smirnov test (KS2) confirms that this difference is statistically significant also accounting for vent position uncertainty (significance level of 0.01, Supplementary Figure 3), confirming the already observed progressive inward migration of post-NYT volcanism (Di Vito et al. 1999; Orsi et al. 2004; Isaia et al. 2009; Rivalta et al. 2019).

### 2.2 Direction of dykes propagation

The direction of propagation of the dykes, which is instead controlled by topographic asymmetries, can be investigated by analyzing the azimuth of the propagation with respect to the centre of the caldera, parameterized as the angle to the North of the line connecting the centre of the caldera and the vent. In Figure 1B, we report the empirical distributions for the different epochs (bins of 20 degrees, the corresponding empirical cumulative distribution functions are reported in Supplementary Figure 2), showing that most of eruptions have an azimuth around 50°, and no specific differences between the distributions





are visible. This observation is tested again with a two-sample Kolmogorov-Smirnov test (s.l. 0.01, see Supplementary
Figure 3), confirming that the directions of dyke propagation are similar in all epochs, with a primary peak toward NE (50°,
toward Astroni, Agnano, and Solfatara).

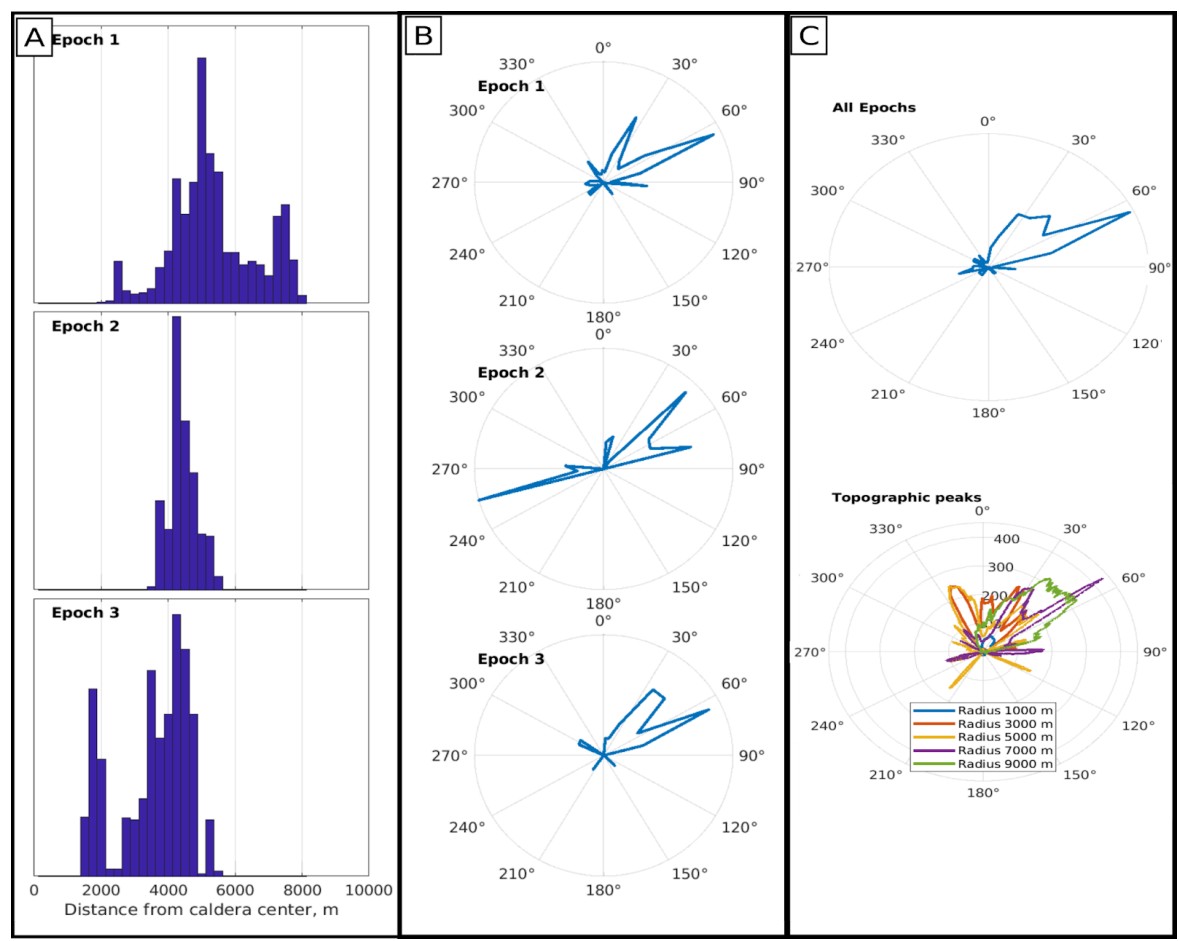


**Figure 1: (A) Empirical distribution of distances from the center of the caldera (dyke propagation length) for Epochs 1, 2 and 3.**
**(B) Empirical distribution of azimuth (dyke direction) for Epochs 1, 2 and 3. (C) Empirical distribution of azimuth (dyke**
**direction) for all Epochs and azimuth of topographic peaks.**

Rivalta et al. (2019) suggest that preferential directions may be induced by topographic peaks that modify the stress field. To
investigate this empirically, we analyse the maxima of the topography surrounding the caldera, retrieving the maxima in all
directions within a given maximum radius. Maximum radii from 2 to 10 km are tested: for each radius, the distribution of
topographic maxima as a function of azimuth is normalised and compared with the azimuthal distribution of past eruptions
(Figure 1C). For simplicity, the present day topography is adopted, even if some of the edifices (and the corresponding
topographic peaks) were built during the post-NYT activity. The comparison confirms the correlation anticipated by Rivalta
et al. (2019). The primary peak around 50° corresponds to the topographic peak associated with Starza (for a radius of 1 km)





and Camaldoli (for radii of 3 km and larger). In agreement with Rivalta et al. (2019), the latter topographic peak is the most
pronounced and it coincides with the highest concentration of eruptive vents during various eruptive periods. A secondary
appears in the NNW direction (-45°) for radii larger than 5 km, corresponding to the peak of the Gauro volcanic edifice
(Monte Barbaro): this edifice was created during one of the first eruptions of the first epoch, and does not correspond to any
peak in the observed distribution of azimuth. Performing a Kolmogorov-Smirnov test between the angular distributions of
past vents and of topographic peaks, the largest p-values correspond to a maximum distance of 7 km, which is the only case
for which the null hypothesis of equal distribution is consistently not rejected independently from vent position uncertainty
(s.l. 0.01, Supplementary Figure 4).
**3 Vent opening probability**
The empirical distributions obtained in Section 2 can be used to set a vent opening probability map. The probability density
function in a specific point in the caldera can be calculated in polar coordinates using the following formula:
$$f_{pol}(r, \theta) = f_r(r) f_\theta(\theta). \tag{1}$$

where the term $f_\theta(\theta)$ is the angular probability distribution, the term $f_r(r)$ is the probability distribution for the distances
from the center of the caldera. In this formulation, it is assumed that these two distributions can be considered independent,
being the latter fundamentally controlled by the nearly circular shape of the caldera and the former predominantly controlled
by local topographic features. This probability distribution can be transformed into Cartesian coordinates as follows:

$$f_{xy}(x, y) = \frac{1}{r} f_{pol}(r, \theta) = \frac{1}{r} f_r(r) f_\theta(\theta) \quad . \tag{2}$$

where the term $f_{pol}(r, \theta)$ is factorized in its two independent terms.
To develop the probability map, we define a grid 14.5 x 12.5 km, centered at LON 427406 and LAT 4518958 (UTM WGS
84 33N), with 700 square cells 500 x 500 m, equal to the one adopted in Selva et al. (2012). Then, the probability in each
cell can be computed by numerically integrating $f_{xy}(x, y)$ computed through Eq. (2), where the terms $f_r(r)$ and $f_\theta(\theta)$ can
be set using the empirical distributions developed in Section 2.
In particular, we propose 2 alternative implementations. At first, we consider for both $f_r(r)$ and $f_\theta(\theta)$ the empirical
distributions that may be considered representatives of the present state of the caldera. In particular, we set $f_r(r)$ as the radial
distribution of the eruptions of Epoch 3 (including also Monte Nuovo), which is indeed the most recent and significantly
different from the one of the previous epochs. Then, we set $f_\theta(\theta)$ as the azimuthal distribution of all post-NYT eruptions, as
the different epochs are statistically indistinguishable. The result is reported in Figure 2A. A second implementation is also
tested by substituting the empirical azimuthal distribution with the distribution of the topographic maxima for a radius of




7,000 m, the one that better correlates with past vents. The results are reported in Figure 2B. The numerical values for both
models are reported as Supplementary File.

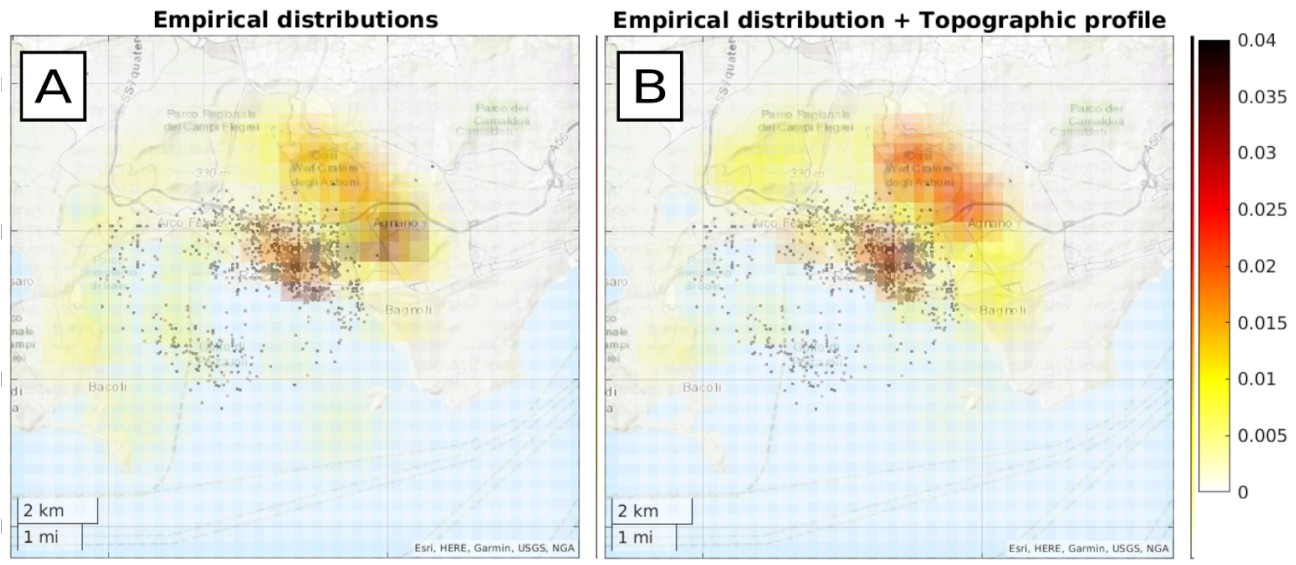


**Figure 2: Vent opening probability maps: (A) based empirical distances and azimuth, (B) based on empirical distances and**
**topographic azimuth. Black dots indicate seismicity (Md > 1.0) in the period June 2023 - June 2025.**

## 142 4 Discussion

The two probability maps in Figure 2 are similar, with two distinct probability peaks in the NE direction at about 4 and 2 km
from the center, corresponding to the Agnano-Astroni and the Solfatara area, respectively. Considering the topographic
contribution (Figure 3B), the angular probability values are more distributed than when the empirical distribution is
considered (Figure 3A). The effect is that the maximum probability value in the area at NW is almost halved. The relative
peaks in the other directions appear relatively more evident when only empirical distributions are considered, generating two
concentric and separated rings at about 2 and 4 km, with secondary peaks also in the submerged side of the caldera toward
the E and SE, in the direction of Baia and Bacoli, as well as the inland area toward NNW.
Comparing these results with the main probability maps for Campi Flegrei discussed in the literature, interesting
coincidences and some significant differences emerge. The map produced by Alberico et al. (2002) differs widely from all
others, having maximum values in the center of the caldera and close to 0 in the peripheral areas, which is in contrast with
the empirical evidence. The higher probability areas of Selva et al. (2012) and Bevilacqua et al (2015) (Figure 3A-B), such
as the Agnano area to the NE of the caldera, coincide with those identified here. The two peaks at distances of 2 and 4 km
from the caldera center are in the range forecasted by Rivalta et al. (2019) when adopting only Epoch III data (Figure 3C).
All these salient features are indeed present in all studies, as they essentially correspond to the spatial distribution of past
vents. However, there are also some significant differences. The propagation distance distribution here produces two distinct



peaks at 2 and 4 km, more than a continuous distribution in this range, generating two distinct rings at relatively higher
probability. In the areas of Nisida and Campo Miseno, the probability values obtained here are relatively lower than in
literature studies. The secondary peak identified in the Averno area in Selva et al., (2012) appears only in the map based on
topography, but it is significantly shifted eastward, in the direction of the Gauro, due to its topographical peak.

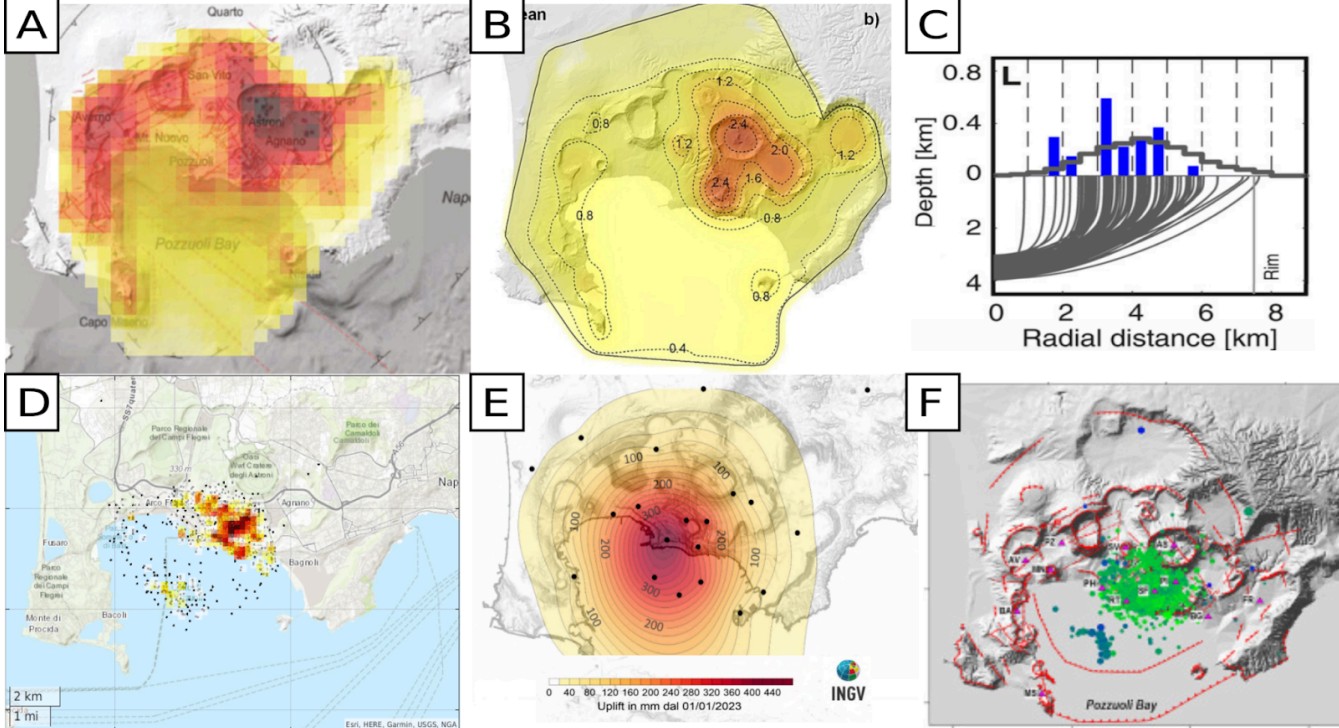

**Figure 3: (A) Vent opening probability maps form Selva et al. (2012). (B) Vent opening probability map from Bevilacqua et al. (2015). (C) Forecasted distances from the center of the caldera from Rivalta et al. (2019) (D) Spatial density of Campi Flegrei seismicity (Md > 1) in the period June 2023 - June 2025. (E) uplift in the period January 2023 - June 2025 (modified from Osservatorio Vesuviano 2025). (F) 2011–2022 seismicity (green circles) and main structural features (magenta lines): outcropping faults, the Campanian Ignimbrite and Neapolitan Yellow Tuff caldera borders and outer/inner rims (modified from Tizzani et al 2024).**

Also the probability peak at Solfatara, which is found in both maps produced here, is more evident here than in literature
studies. This is a consequence of combining the peaks in distance and angles. This effect is also evident in the other
secondary probability peaks at sea in the E and SE directions, in the direction of the topographic peaks of Baia and Capo
Miseno. These peaks emerge here, but they were not found in literature studies, and they essentially coincide with one spot
of high seismic activity recorded in the ongoing unrest episode (Fig. 3D), as well as in the past monitored unrest episodes. In
particular, in Fig. 3D, we report the spatial density of the seismicity recorded in the last two years, which shows a striking
correspondence to the vent probability maps reported in Fig. 2A.



More in general, the two higher probability rings that emerge here correspond to the position of the ring faults surrounding the inner caldera and bordering the area of maximum uplift (Fig 3D and 3F, Tizzani et al. 2024, Natale et al. 2025). These observations are completely independent and may help spotting privileged paths for magma ascent around the NYT caldera border.

**Datasets**

The position of the vents related to past eruptions was obtained from Bevilacqua et al. (2015). The DEM of the area obtained from TINITALY (Tarquini et al. 2023) with WGS 84 / UTM zone 33N coordinates, available at https://tinitaly.pi.ingv.it/.

**Acknowledgements**

The figures and maps have been produced using Matlab and/or InkScape software.

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
