# Peer review of "Brief communication - Vent opening at Campi Flegrei: clues from"

_EGUsphere, 2025_

## Author Comment (AC1)

We thank the reviewer for the very constructive revision. We implemented all the suggestions, and we think that the manuscript has overall improved significantly.

In particular, we thoroughly revised all citations, also thanks to the important suggestions given by the reviewer. We improved the presentation of the method, and we now better specify the effective assumptions behind the model, by specifying both the assumptions of Rivalta et al. (2019) and those assumptions that are inherited in our study. We also significantly updated the discussion, making more quantitative and explicit the comparison with both previous models and the ongoing seismicity.

Regarding the assumptions, as now better discussed in Section 2, we note that they are somehow looser than the one adopted in Rivalta et al. (2019). Indeed, we empirically track potential propagation of dykes from a central area of the caldera toward the position of past vents, but we do not need any assumption about depth and shape of the source (like in Rivalta et al. 2019), as we do not model dyke propagation, but we simply empirically analyze past vent positions. In other words, our model is inspired by the physical process described in Rivalta et al. (2019), which stands for an independence between distance from the center and direction of the propagation of dykes, but we do not adopt any physical modelling here. This very important point is now better described in the manuscript.

Overall, we think that the manuscript significantly improved thanks to this revision. All these improvements are specifically discussed in the attached file.

| Dear Editor, | We thank the reviewer for the very constructive revision. We tried to implement all the suggestions, and we think that the manuscript has overall improved significantly. |
|---|---|
| I have completed a thorough review of the manuscript entitled "Brief Communication – Vent Opening at Campi Flegrei: Clues from Dyke Propagation Patterns" by Selva and Mangone. | We now also significantly improved the discussion, making more quantitative and explicit the comparison with both previous models and the ongoing seismicity (Fig. 3). |
| This work provides a concise overview of the current understanding of vent-opening maps at Campi Flegrei. It introduces a novel empirical methodology that incorporates the spatial distribution of past vent locations along with the influence of topographic relief on dyke trajectories. The results, presented as vent opening probability maps, suggest two | |

concentric annular maxima at approximately 2 km and 4 km radial distances, with a general peak toward the northeast and additional local maxima corresponding to topographic highs. The authors also note a qualitative correspondence between their results and the epicentral distribution of recent seismicity. While this point is briefly mentioned in both the abstract and the discussion, it is not explored in depth, particularly regarding the significant variation in the depth distribution of seismicity across different sectors of the caldera.

Overall, the manuscript is clearly written and logically structured. Nonetheless, there are several areas in which the presentation could be improved, particularly concerning the manuscript's structure, the accuracy and placement of literature citations, the clarity of the figures, and certain aspects of the wording.

| | |
|---|---|
| The proposed approach is both interesting and promising; however, the manuscript would benefit from a more explicit articulation of the underlying assumptions. For instance, the authors adopt a similar assumption to that of Rivalta et al. (2019) regarding the dyke source, While this assumption is consistent with certain modeling efforts, it simplifies the broader volcanological literature, which often supports the existence of a sill-like | Thank you for this comment. We now better specify the effective assumptions behind the model, by specifying both the assumptions of Rivalta et al. (2019) and those assumptions that are inherited in our study.

To this end, we now specify that

*"Rivalta et al. (2019) assumed that the origin at depth of the magma is located at the centre of the caldera, below the location of the maximum observed uplift (Amoruso et al. 2014, Rivalta et al. 2019, Buono et al. 2025), originated at 3 km depth and* |

| | |
|---|---|
| source. Furthermore, the presumption of a fixed source depth of 3 km for all post-NYT eruptions may lack robust petrological justification. The model also assumes predominantly lateral dyke propagation, a simplification that may not hold uniformly throughout the post-NYT eruptive period. It would be beneficial for the authors to state these assumptions more explicitly at the outset. | *producing mostly lateral propagations with trajectories controlled by the structure of the caldera and the consequent stress field. The physical process described in Rivalta et al. (2019) suggests a potential independence between direction of dyke propagation and distance from the caldera center, as they are controlled by two different features of the caldera, potentially leaving an empirical track in past vent positions. Assuming a magma origin located around the center of the caldera, independently of its specific depth and geometry, this empirical track may be retrieved by studying the distribution of past vents around the caldera centre…"* |
| The referencing of literature requires refinement, as several citations appear either outdated or incorrectly positioned—particularly those pertaining to the structural evolution of the caldera and its eruptive history. More recent studies have revised the temporal framework of Campi Flegrei's activity, extending it beyond 200 ka, and have updated interpretations of the caldera structure. For instance, the structural rims are now understood to be primarily associated with the CI eruption and subsequently reactivated during the NYT phase (e.g., Natale et al., 2022, Journal of Structural Geology). Additionally, recent investigations into the physical properties of dykes within the Campi Flegrei system should be acknowledged (e.g., Buono et al., 2025, AGU Advances; Natale and Vitale, 2025, Nature Communications). | The number of references is limited 30 in "Brief Communitations",and this influenced us a bit in the writing. However, we agree with the reviewer and we added several more, as suggested. We hope the Editor will agree with this. |
| In summary, I find the manuscript to be of overall high quality and recommend it for publication pending moderate revisions. | We carefully reviewed and also implemented all line-by-line suggestions. |

| | |
|---|---|
| Please find below line-by-line minor suggestions and comments.

Best regards | |
| [...] Line 9: Consider adding the term "azimuth" to clarify the reference to the orientation of the dykes. | Added |
| Line 10: The two principal peaks should be more clearly emphasized in Figure 2, which is currently difficult to interpret. | We modified Figure 2, and now the peaks are more evident. |
| [...] Line 14: The earliest volcanic activity observed in outcrops within and beyond the caldera is estimated at approximately 80 kyr, based on Pappalardo et al. (1999) and Scarpati et al. (2013). However, recent studies of widespread tephra layers in Italy and the Mediterranean extend the chronology of CF activity to nearly 200 kyr (e.g., Monaco et al., 2022 - GPC; Fernandez et al., 2024 - QSR). This includes a recently identified large-magnitude eruption at 109 ka (e.g., Fernandez et al., 2025 – Communications Earth and Environment). Tephra deposits of comparable age and composition have also been retrieved from boreholes (e.g., Sparice et al., 2024 – JVGR). These more recent contributions should be acknowledged. | Thank you for this comment. We corrected the text accordingly, and we added the references. |
| Line 15: Please include a reference to the Campanian Ignimbrite eruption/age. | Added |
| Line 15: The statement that all | We removed the sentence, as not |

| | |
|---|---|
| eruptions occurred within the CI caldera rim is not accurate. For example, eruptions took place at Procida Island (De Astis et al., 2004) and at CFc, such as the Torregaveta eruption. | necessary. |
| Line 17: The citation of Sbrana et al. (2021) is inaccurate, as these authors attribute the caldera structure solely to the CI eruption. | We removed the sentence, as not necessary. |
| Line 18: The term "inner caldera" has a specific structural meaning, yet many post-NYT vents are located beyond this boundary. Orsi et al. (2004) is incorrectly referenced here, as it does not address volcano-tectonic structures. | We removed the word 'inner', as not necessary. |
| Line 19: A citation to Di Vito et al. (1999) would strengthen this statement. | Added, thanks |
| Line 19: Consider modifying the sentence to read "comprises at least 33 eruptions." | Corrected, thanks |
| Line 19: Correct "spaning" to "spanning." | Corrected, thanks |
| Line 20: Adjust 15,000 to 14,000 for consistency with line 16. | Corrected, thanks |
| Line 22: Revise 9.200 to 9.100. | Corrected, thanks |
| Line 23: Revise 28 to 26. | Corrected, thanks |
| Lines 23–25: Consider dividing the sentence after "3,800 years ago." A | Corrected and integrated, as suggested |

| | |
|---|---|
| more precise description could be: "Its activity was predominantly concentrated in the northeastern part of the caldera (Agnano area), secondarily in the northwestern sector (Averno area), and concluded with peripheral distal eruptions (Nisida, Capo Miseno, and Fossa Lupara)." Recent works provide additional context (Natale et al., 2025 – GSA Bulletin). | |
| Line 27: Update the reference to Di Vito et al. (1987 – Bulletin of Volcanology). | To avoid too many additions, we added directly Di Vito et al. 2016 (Scientific Report) |
| Line 28: It would be useful to cite recent studies on caldera resurgence at La Starza cliff (Isaia et al., 2019 – JVGR; Natale et al., 2022 – Basin Research). | We added the suggested citations, thanks |
| Line 31: The history of ground deformation since 35 BCE has been reconstructed by Di Vito et al. (2016 – Scientific Reports). Vitale and Natale (2023 – Earth, Planets, and Space) describe the long-term deformation pattern. | We added the suggested citations |
| Line 33: Correct the date to 1982–84. | Corrected, thanks |
| Lines 34–35: Suggested rephrasing: "which fully recovered the subsidence in 2021, and now exceeds the uplift peaks observed in the last century." If this statement refers exclusively to deformation, cite Bevilacqua et al. (2024). If it encompasses seismicity and degassing, please include additional references accordingly. | We modified the text accordingly. |

| | |
|---|---|
| Line 40: Suggested revision: "[...] parameters, focusing on the tectonic structures recognized at that time, to track [...]." | Corrected, thanks |
| Line 45: Update to Charlton et al. (2020), not 2018. | Corrected, thanks |
| Line 46: Replace with "corresponding to the caldera rim." | Corrected, thanks |
| Line 52: Suggested rephrasing: "Rivalta et al. (2019) analyzed the effects of caldera unloading, as well as those of topographic peaks, [...]." | Corrected, thanks |
| Line 55: Insert "may": "[...] creates a stress field that may favor magma trajectories [...]." | Corrected, thanks |
| Line 56: Add "(unloading effect)": "[...] of the caldera (unloading effect) significantly [...]." | Corrected, thanks |
| Line 57: Clarify the intended wording: "Geometric centre at a given distance"? | We mean 'geometric centre of the caldera'. Corrected, thanks. |
| Line 60: Replace with "left a trace." | Corrected, thanks |
| Line 66: Correct citation to Amoruso et al. (2014). | Corrected, thanks |
| Line 67: Refer also to the general comment above regarding assumptions on the dyke source. | As suggested, we added here the assumptions, as discussed above. |
| Line 68: Verify coordinates, as longitude (Easting) appears to contain one digit | Corrected, thanks |

| | |
|---|---|
| too many, while latitude (Northing) appears to be missing one. | |
| Line 76: Please ensure consistent use of terminology ("dyke" vs. "dike") throughout the manuscript. | Corrected, thanks |
| Line 85: Among the cited works, only Rivalta et al. (2019) appear to state this explicitly. | Corrected, thanks |
| Line 88: The assertion regarding topographic control seems to be treated as a fact, whereas it should be presented as a working hypothesis. | Now the assumption is better stated in the previous section, and it is recalled here as "Under the assumption of propagation from the caldera center, this distance represents the length of the horizontal propagation of the dykes that alimented such eruptions." |
| Line 101: Consider expanding the statement: "[…] the stress field is mainly controlled by unloading." | Added |
| Line 102: Please clarify whether Figure 1c represents simple topographic profiles or averaged swath profiles. The latter would provide a more representative average topography. | They are the maxima in radial swath profiles with specific length. Now we added this in the caption and in the main text. |
| Line 107: Use "La Starza marine terrace." | Corrected, thanks |
| Line 107: Clarify orientation: If La Starza terrace is located NW of the center, it should correspond to ~340° azimuth, not 50°. | La Starza is at NE of the caldera center (as now specified in the text), thus it is at around 50°. |
| Line 108: Correct to "Camaldoli Hill." General: These locations should be shown on a map, as readers unfamiliar with Campi Flegrei may otherwise be | Corrected, thanks |

| | |
|---|---|
| confused. | |
| Line 112: This observation indicates an anticorrelation, or at least underscores the need to state at the outset that the spatial distribution of vents is hypothesized to be controlled by topography. | The peak toward NNW is present only for small radii, up to 5 km, while it is almost negligible for larger radii. This is in agreement with the results of the test that we describe in the following lines. Now we made this point clear in the text. |
| Line 120: Invert the order of this sentence and the next, so that it follows the order presented in the formula. | Done |
| General comment on Figure 2: Figures could be made more legible by using a lighter, more uniform background (e.g., shaded relief). The seismic hypocentre dots currently obscure the NE peak within the inner circle. | We followed this suggestion, and indeed Figure 2 improved significantly. |
| Line 142: Consider renaming this section "Discussion and Conclusions," as the final sentences are conclusive in nature and a separate Conclusions section is absent. | Done |
| Line 149: The reference to "E and SE" is unclear. Does it pertain to Baia and Bacoli, or to Bagnoli and Nisida? Please clarify and specify locations. | The direction is W and SW referred to Baia and Bacoli. |
| Line 151: It would be appropriate to include the map from Alberico et al. (2002) to illustrate similarities and differences, rather than requiring the reader to consult that paper independently. | We now removed all the original maps and we report the map of the differences, including Albertico et al. (2002). In particular, we added the comparison in Fig. 3A,D and the original maps in Supplementary Figure 7. |
| Line 170: The difference in the Solfatara peak is not particularly pronounced; it | We now made a specific and more quantitative comparison with both models |

| | |
|---|---|
| is already visible in Selva et al. (2012) and Bevilacqua et al. (2016). | (new Fig. 3), making the comparison much more explicit. This comparison highlights that in the Solfatara area, the peaks of the models developed here are more pronounced than in all previous studies, especially for model M1 (empirical without topography) |
| Comment on Figure 3F: According to the caption in Tizzani et al. (2024) and cited references, the rims shown in Tizzani et al. (2024) correspond to the CFc caldera rims, not the CI and NYT rims. Revising accordingly would improve consistency throughout the text. | Corrected, thanks |
| Line 172: The directions are unclear. By "E and SE," do you, in fact, mean "W and SW," referring to the topographic highs at Baia and Capo Miseno? | Yes, W and SW. Corrected, thanks. |
| Line 178: Natale et al. (2025) is not included in the reference list. | Added, thanks. |
| Final comment: A dedicated "Concluding Remarks" section is missing and should be provided if the name of the section is not updated as suggested above. | We updated the section's name as suggested. |
| I trust that these comments will assist the authors in enhancing the clarity, accuracy, and overall quality of their manuscript. | They helped improve the text a lot indeed. Thank you for this very constructive revision |

---

## Author Comment (AC2)

We thank the reviewer for this revision and the interesting suggestions provided. We essentially implemented all the suggested analyses as specified below, significantly improving the manuscript.

In particular, we extended the text to better describe the methods and related assumptions (Section 3), adding several new analyses and leading to an update of all figures and to the addition of 3 supplementary figures. We now split methods and results in two sections (new section 4). We significantly extended the discussion, by adding all the suggested additional points. We added a more quantitative comparison between our results and all the previous models existing in literature, as well as with the other geophysical data. We also added a concise conclusion session (section 5), which includes a specific discussion about epistemic uncertainty on vent opening, as detailed below. These changes address all reviewer's suggestions.

Among the added analyses, we now specifically study the the correlation between the radial and azimuthal components in past data, as well as we implemented the suggested kernel for the analysis of empirical distributions, Regarding specifically the independence between radial and azimuthal components, we formally tested the hypothesis of independence through standard hypothesis testing (Section 2.2, Supplementary Figure 5). The results stand for the independence that was earlier (in the first version of the manuscript) assumed only based on physical considerations (see comments below for more details). This test is now described in the main text (end of Section 2), and an additional supplementary figure was produced to show the test results in detail (supplementary figure 5).

Regarding uncertainty quantification, we agree that it is an important topic. As now highlighted in the discussion, our model "... *may provide the possibility to better constrain the epistemic uncertainty about vent opening probability. Indeed, several case studies recently demonstrated that the effective epistemic uncertainty on a target physical process (here vent opening) is better estimated by combining alternative approaches than by exploring the epistemic uncertainty inherent to individual models (Selva et al. 2015; Marzocchi et al. 2021; Meletti et al. 2021, among the others), defining weighted ensembles of existing models (SSHAC 1997, Marzocchi et al. 2017). This quantification may be the topic for future works, and the approach presented here may be a significant added value to this end, by providing an effectively alternative approach to vent opening probability quantification." (Section 5).* In other words, here we concentrate on describing one approach that is effectively an alternative to the ones available in literature. The very existence of a completely different approach is an added value for future epistemic uncertainty quantification. However, in this manuscript we prefer avoiding the additional complication of quantifying an epistemic uncertainty on the specific approach that surely represents an underestimation of the effective epistemic uncertainty on the process (vent opening). This very important discussion is now added in the conclusion, as the last point of the manuscript.

Overall, we thank the reviewer for the important comments that gave us the opportunity to significantly improve the manuscript. All these improvements are specifically discussed in the attached file.

| The manuscript describes an interesting approach to define a vent opening map at Campi Flegrei caldera | Thank you for this revision and the interesting suggestions provided. We essentially implemented all the suggested analyses, significantly improving the manuscript. We significantly extended the |
| --- | --- |

| | |
|---|---|
| by splitting the density function in polar coordinates as the product of radial and azimuthal components. While this formulation is promising in studying the vent opening patterns, the manuscript does not provide any convincing argument for assuming independence between the radial and the azimuthal patterns, nor an uncertainty quantification of the output. In fact, splitting the density function in two factors is equivalent to assume they are independent, and that seems a pretty strong assumption, to me, although it may provide an interesting point of view on the problem.

The manuscript, after a general introduction, follows with a description of the empirical distributions of direction and radial distances of past vents from the caldera center. Then, it describes the split formulation of vent opening probability, before a Discussion section in which the new vent opening maps are briefly described and compared to some pre-existing vent opening maps and to the current geophysical unrest pattern. | text to describe the new analyses, updating all figures and also adding 3 supplementary figures. We also significantly extended the discussion, as suggested.

Regarding the independence between radial and azimuthal components, we tested the independence in existing data (Section 2.2, Supplementary Figure 5), confirming the independence that was earlier assumed only based on physical considerations (see comments below for more details). We now added the description of the test and an additional supplementary figure to show the results. |
| The manuscript showcases interesting ideas, but it is incomplete and unconvincing in providing a new vent opening map. Also, the methodology section is missing and the methods are mixed with the results section, and insufficiently detailed. Another problem is the unclear comparison | We significantly edited the manuscript, also thanks to reviewers' comments. In particular, we better clarified the main assumptions, and we improved the presentation of the method.

We separated methods from results and discussion (now sections 3 and 4), and we added a concise conclusion section (section 5). In section 4, we improved the comparison with previous methods and with geophysical data (new Fig. 3), by reporting |

| | |
|---|---|
| between past vent opening positions and geophysical unrest patterns, in the discussion.

The text is generally well written, but I am not a native speaker so I may have overseen minor language issues. The key novelty of this research stays in the split formulation, but that would deserve a more extensive analysis and discussion. | the suggested analysis of the difference with the previous methods (we also added the comparison with Alberti et al., as suggested by reviewer 1), and by producing specific maps to compare earthquakes and structural features. |
| In summary, the results are incomplete and the discussion section is disappointing. I strongly suggest re-shaping the manuscript by adding new analyses, separate the methods from results, and rewrite the Discussion. Probably, a plot of the differences between the pre-existing vent opening maps and the results of the manuscript would help to highlight how the previously published maps differ from the manuscript's results. Very importantly, if the authors aim at introducing a new vent opening model, they should provide an uncertainty quantification of the probability density values. Also, having a clear and concise Conclusion section would be really appreciated. Finally, please consider if a "plain" research article would not be a better format for this manuscript. | We significantly improved the discussion section, adding most of the reviewer's suggestions. We added new analyses like the correlation analysis and test, we introduced a kernel for the analysis of empirical distributions, we made a quantitative comparison between this model with all models existing in literature, and we improved the comparison with the other geophysical data (Fig.3). We separated methods and results, and we significantly improved the discussion section, addressing all reviewer's suggestions. All these improvements are specifically discussed below.

As now discussed in the Conclusions (Section 5), regarding the epistemic uncertainty quantification, we agree that it is an important topic. As now highlighted in the discussion, our model "... *may provide the possibility to better constrain the epistemic uncertainty about vent opening probability. Indeed, several case studies recently demonstrated that the effective epistemic uncertainty on a target physical process (here vent opening) is better estimated by combining alternative approaches than by exploring the epistemic uncertainty inherent to individual models (Selva et al. 2015; Marzocchi et al. 2021; Meletti et al. 2021, among the others), defining weighted ensembles of existing models (SSHAC 1997, Marzocchi et al. 2017). This quantification may be the topic for future works, and the approach presented here may be a significant added* |

| | |
|---|---|
| | *value to this end, by providing an effectively alternative approach to vent opening probability quantification.".* In other words, here we concentrate on describing one approach that is effectively an alternative to the ones available in literature. The very existence of a completely different approach is an added value for future epistemic uncertainty quantification. However, in this manuscript we prefer avoiding the additional complication of quantifying an epistemic uncertainty on the approach that surely represents an underestimation of the effective epistemic uncertainty on the process (vent opening). This very important discussion is now added in the conclusion, as the last point of the manuscript. |
| L21 – please add a reference for the 12300 BP age, in fact the 2022 book by Orsi et al. reports 11900 to 12200 BP. | We added the reference to Bevilacqua et al. 2016, thanks. Here this detail is not used in the paper, so we treat this as a general indication, without discussing details about it. For this reason, we just rounded all numbers indicating averages. |
| L24 – when you say that the third epoch is predominantly focused in the NE part of the caldera and secondarily in the NW sectors, you should also mention the Nisida and Capo Miseno exceptions. | Added, thanks. |
| L25 – is the 4550 BP age an average of 4500 to 4600 BP? Add reference please. | Yes, we averaged. We now reported 4500 and we added the reference to Bevilacqua et al. 2016. Again, this detail is not used in the paper, so we treat this as a general indication, without discussing details about it. |
| L29 – "geometrical center" is unclear. | We reformulated in order to make more clear this sentence |
| L33 – why 1984-1986? Should not be 1983-1985? | Yes, there was a mistake. Usually this unrest is referred to as 1982-84, as in the cited Del Gaudio et al. 2010. Now, we corrected the text. |

| | |
|---|---|
| L40-44 – please mention that both Selva et al. 2012 and Bevilacqua et al. 2015 are vent opening maps with uncertainty quantification, i.e., the probability density values have an uncertainty distribution. This is a key point. | Added, thanks. |
| L62 – Why you speak of past dykes? I would find much clearer if you spoke of past vents distances. In fact, we don't really know if all (or any) dykes actually started from the caldera center. | We updated the title, as suggested. In any case, we now better specify the assumptions of our study, which are similar to the ones of Rivalta et al 2019, who assumed that the dyke propagation starts approximately at the center of the caldera. |
| L74-75 – Is 1000 samples the most appropriate sample size? Did you try with more, or less, samples? What changes? Also, by looking at the Supporting File I see that you probably used a uniform distribution inside the elliptical shapes of Bevilacqua et al. 2015; I can't find the uniform distribution specified anywhere, tough. | The samples are used to check the stability of all statistical analysis and tests. All cumulative distributions shown in Appendix indicate that this number is sufficiently large to this aim. Regarding the distribution's shape, yes, you are right, they are uniformly distributed. This is what we meant by "randomly selected within bounds". Now we made it more explicit in the text, saying that they are "uniformly distributed". Thanks. |
| L76, L87 – again, please do not speak about dykes. It is unrequired to assume that all dykes propagated from the caldera center if you just speak about vent opening patterns. It is ok to speak about dykes in the discussion, but it is not necessary here. | We modified accordingly |
| L119 – This formulation is not valid for every 2D density function. You can do that if (and only if) the two components (radial and azimuthal) are independent. This is really a key point that you missed to discuss. | Thank you for this comment. The need for independence was already specified in line 121 of the original text, and it was assumed because of the independence of the physical processes leading to different directions and distances (being the distance fundamentally controlled by the nearly circular shape of the caldera and the direction predominantly controlled by local topographic features). |
| | However, we did not test it explicitly, and we |

| | now added this test (Section 2.2). The results (discussed in Section 2.2 and reported in Supplementary Figure 5) show that the hypothesis of equal distribution cannot be rejected for all combinations of sectors, standing for the independence of the two parameters. |
|---|---|
| L140 – Please delete the earthquakes from Figure 2. They do not belong to the vent opening pattern analysis that you are describing here and they hide the probability density function values. Please add contour lines. The comparison of vent opening map and seismic patterns could be done in the discussion section, once it is clarified its meaning. | We modified the figure accordingly, removing the background to better highlight the probability values. We prefer to avoid contourlines, as the computations are made over a grid and we prefer that this appears in the figures. |
| L142 – The result section is unfortunately incomplete. Some examples of required additions are listed below. | Thank you for the suggestions. We tried to improve this section, also addressing all the following points. |
| 1. You assumed that azimuthal and radial distributions are independent. But how the bivariate plots of distance and direction of past vents looked like? If they looked much correlated, it should be a concern for the validity of this formulation in producing a new vent opening map. However, this analysis could be useful anyway to understand that correlations in polar coordinates may play a role in shaping the vent opening patterns. This should be discussed. | We now report the bivariate plot, along with the results of the independence test discussed above, in supplementary figure 5. |

| | |
|---|---|
| 2. Why you did not try to produce maps based on the three epochs? I can't see why you just tested the third epoch after having analyzed the marginal distributions of the three epochs – by doing that, you discarded half of the vent opening dataset. | We discharged half of the events simply because we demonstrate that Epochs 1 and 2 are significantly different from Epoch 3. Thus, their introduction may introduce a bias. This is now explicitly commented on in Section 3, just after equation 2.

Notably, this is in agreement with Orsi et al. (2004) that concluded that the last change in stress regime occurred prior to onset of the Epoch 3, suggesting that only the past 5 ka should be considered as reference for the present state of the caldera. For this reason, Epoch 3 was taken as reference also in Orsi et al. (2009) and Selva et al. (2012). Also this is now discussed at the beginning of section 4. |
| 3. How the differences between E and W sectors could find integration in this analysis? I believe this distinction was important in shaping some of the pre-existing vent opening maps. Please give a look over 10.3389/feart.2017.00072. | The difference between E and W sectors discussed in that paper does not pertain to vent opening, but more in the size of eruptions (and specifically in PDC areal distributions). The size may be influenced by multiple factors, and to discuss this is out of the scope of this manuscript. |
| 4. Why you used empirical distributions and not kernel functions? Is this changing much if you assume a simple Gaussian kernel with an appropriate bandwidth? | Thanks for this suggestion. We now implemented kernel functions to smooth the distributions, as suggested. The results are practically the same. The implementation and the results are discussed in Section 3, Fig. 1, and in Supplementary Figure 6. |
| 5. How the new formulation deals with a uniformly distributed layer that both Selva et al. 2012 and Bevilacqua et al. 2015 assumed inside the caldera? | This assumption is not made here. The size here is empirically controlled by the maximum distance recorded in the past, which is about 8 km from the caldera center. here, as the distribution of distances never go beyond caldera borders. We now specifically comment on this just above Fig 3. |
| 6. Could you try quantifying the uncertainty affecting all this? You may do that as you wish, but you | The uncertainty on vent position is treated in all tests. Regarding the epistemic uncertainty on the vent opening probability, as we now comment on the paper (last part |

| | |
|---|---|
| should not totally oversee that step. | of the discussion):

 *"Overall, the maps proposed here are based on assumptions radically different from the ones discussed in literature. This may provide the possibility to better constrain the epistemic uncertainty about vent opening probability by producing an ensemble of alternative models, in which all models consistent with data are combined to provide an ensemble model (SSHAC 1997, Marzocchi et al. 2017). Several case studies recently demonstrated that the effective epistemic uncertainty on the target physical process (here vent opening) is better estimated by combining alternative approaches than by exploring the epistemic uncertainty inherent to individual models (Selva et al. 2015; Marzocchi et al. 2021; Meletti et al. 2021, among the others). Thus, uncertainty on vent opening probability in Campi Flegrei may be may estimated by defining an ensemble of all available models, including the ones produced here as well as all previous models available in literature, along with a proper quantification of potential credibility weights (SSHAC 1997; Marzocchi et al. 2015): this may be the topic for future works. This paper may be a significant added value in this sense by providing an effectively alternative approach to vent opening probability quantification."* |
| L150-L155 – I would have liked to see a plot of the differences between the new map(s) and the pre-existing maps (e.g., by considering their mean values). | Thank you for this comment. We added these plots in Figure 3, and we updated the discussion accordingly (Section 4). |
| L158 – I can't see the rings very well in Figure 2, because of the earthquakes plotted on top of it. Please delete them. | We modified Figure 2, removing earthquakes and changing the background, in order to enhance the vision of the probability values. The rings are also now very evident in Fig. 3, when the model developed here is compared with previous studies. |
| L164 – Figure 3: I find this Figure a bit confusing. These are all Figures | We modified this figure. The first figures are now about the differences between alternative maps, as suggested above, |

| | |
|---|---|
| published in other papers, and I can't do a quantitative comparison of these heterogeneous data to the new maps (which are also plotted elsewhere, in Figure 2). Also, the data in plots d-e-f are not about vent opening information. Why mix apples with pears? | extending to Alberico et al. 2002.

We then added two selected maps that are useful for the discussion, and enable a more quantitative comparison between our vent opening models and the distribution of the seismicity and the caldera ring faults. |
| L174-175 – "striking correspondence" is a qualitative claim. Please make it more quantitative. In particular, please explain how a comparison between your vent opening map and seismic pattern should be read. In general, main active faults are certainly correlated to seismicity, and main active faults are also hypothesized to be correlated to vent locations in several of the pre-existing vent opening maps. What is really the point of this sentence? Is it pointing at the possibility of new vent opening models also integrating seismicity? Or is it somewhat trying validate the current vent opening maps by using geophysical data? | Here, with striking we mean "unexpected". We now reformulated this sentence, along with all this discussion, to make it more clear. We also modified Fig. 3H,I, in order to facilitate a quantitative comparison.

As for the interpretation, we think that whatever interpretation is very speculative at this point. The interesting thing is this correspondence, that may be better understood with future studies. |